# Hildegard of Bingen: Philosophical Life and Spirituality

Peter Harteloh

Erasmus Institute for Philosophical Practice, 3032 AD Rotterdam, The Netherlands; info@filosofischepraktijk.com

**Abstract:** Hildegard of Bingen (1098–1179) was a medieval mystic. From a young age, she had many colorful visions and became well known and influential not only in her own time but in ours as well. Her music reached the mellow house scene in the 1990s, reviving Hildegard's spirituality for people today. In this paper, I will approach Hildegard as a philosophical practitioner and conduct an imaginary philosophical consultation. I will study her biography, listen to her words by some authentic text fragments and propose a spiritual exercise on her music in order not to just think *about* Hildegard of Bingen but to try and think *like* Hildegard of Bingen, in line with the principles of philosophical practice. This way, I will try to understand Hildegard in a practical way and not (just) annotate the regular (theoretical) interpretations of her life. I will distinguish three phases in her life as movements towards spirituality: (1) her relationship with the world, (2) her relationship with God, and (3) her relationship with herself as a spiritual being. I will argue that her life is an example of a philosophical life. Hildegard's "not fitting in any place" (being átopos) and her development define such a life as a path towards an authentic self, attained by spirituality. The paper intends to contribute both to the understanding of philosophical consultations and to the understanding of Hildegard of Bingen.

**Keywords:** Hildegard of Bingen; philosophical life; philosophical practice; philosophical counseling; spirituality

## 1. Introduction

Hildegard of Bingen (1098–1179) was a German mystic who lived in the Middle Ages in monasteries along the river Nahe. Hildegard was born as a high-class German lady. She was gifted with visions from early on, making her differ from the other children of her age. She was sent by her parents to be educated as an anchoress and lead a secluded life far from society. However, she did not fit in with that way of living. She founded her own monastery and became famous as a result of her visions. She attracted many women, who joined her in the monastery, and corresponded with noblemen and popes. In the end, she was not a secluded anchoress, but a mystic, well known and influential not just in her own time but even today. The music she composed enjoyed a revival in the mellow house scene of the 1990s, a way for contemporary people to participate in her spirituality in a practical way.

In this paper, I want to approach Hildegard as philosophical practitioner and conduct an imaginary philosophical consultation. The aim is to contribute to the understanding of Hildegard of Bingen on the one hand and to demonstrate the principles of philosophical consultations on the other. Philosophical counseling is a relatively new phenomenon. Although psychologists already applied philosophy in their consultations, it was the German philosopher Gerd Achenbach who in 1981 started offering individual consultations. It was the beginning of a new paradigm in philosophy. Philosophers followed Achenbach's example and today a philosophical consultation has become a professional interaction between a philosopher and a client (called a visitor by Achenbach) on themes in life with philosophical means (Achenbach 2010). Questioning, interpreting and understanding are the building blocks of a philosophical consultation in order to reveal fundamental concepts

underlying the thoughts of the client and facilitate a change towards autonomy and equilibrium of mind (Harteloh 2023a). So, let's imagine Hildegard visiting a philosophical practice for consultation.[1] Let's disregard the usual or "official" interpretations of Hildegard's life, study her biography, listen to her words and music, and do an exercise to understand her in a practical way.[2]

### 2. The Life of Hildegard

In philosophical counselling, I try to discover themes in the life of a person and provide them with a philosophical interpretation (Harteloh 2013). So, what kind of themes emerge in the life of Hildegard of Bingen? For this, I must first look at her biography (Hildegard of Bingen 2010). Hildegard of Bingen was born in 1098 of noble parentage at Böckelheim (Germany). She started to have visions of luminous objects at the age of three. At the age of 8, the family sent her to an anchoress named Jutta to receive a religious education. She took care of Jutta and was educated by her within the walls of Benedictine houses. In 1112, she entered the isolated convent of Disibodenberg, where she revealed her visions and rose to be abbess. In 1150, she and some of her nuns migrated to a new convent on the Rupertsberg, a finely placed site high in the hills along the river Nahe. Here, Hildegard passed the main portion of her life as abbess of the monastery. She wrote three books on her visions and a mystery play. She conducted music (Burkholder et al. 2010) and corresponded with noblemen, popes and Bernard of Clairvaux (Baird and Ehrman 1995). She died in the monastery of Rupertsberg on 17 September 1179.

As a philosophical counselor, my first impression is that of a person struggling with herself and with the social environment. There is something extraordinary, exhibiting itself early in life, hard to handle for the person herself and for her parents, teachers, fellow sisters or society, who first tried to confine and isolate Hildegard in an attempt to control the extraordinary. However, this attempt did not produce the intended results. The extraordinary did not fit in the confinement of the monastery either and attracted people in such a strong way that they tended to move towards it and be led by it. Hildegard established her own monastery, became the leader of a spiritual movement and was able to create her own world, where she could express herself by writing or music. She settled at a symbolic place, high upon the hills, with an ancient bridge between the Rupertsberg monastery (her world) and Bingen (the world). Hildegard's life is the expression of a development towards a very strong form of autonomy (Lerius 2018), a creation of your own world, attained by spirituality.

In Hildegard's life, three phases can be distinguished: (1) her education and relationship with the world during her teens, (2) her life as a sister at the monastery of Disibodenberg, establishing a relationship with God, and (3) her life as abbess of her own monastery at the Rupertsberg as a matured woman, where she established a relationship with herself as a spiritual being. They represent themes superimposed on each other. Altogether, they constitute three life stages as a movement towards spirituality. This development represents: (1) a so-called horizontal axis on which the relationship of a person with others in the surrounding world is situated, (2) a vertical axis on which the relationship of a person to "The Higher" is situated (e.g., God, fame, quality, justice, etc.), and (3) a reflexive axis which points from the origin of the coordinate system back to the origin, symbolizing the relationship of the person with herself. The localization of a person in this three-dimensional space of meaning serves a systematic exploration of life in philosophical practice.

### 3. Hildegard and Her Relationship with the World

From a young age, Hildegard struggled with her position in the world. She was said to have had visions since the age of three (Hildegard of Bingen 2000). These visions provided her with knowledge that did not match the regular pattern of a life course. Life in the Middle Ages was guided by a simple rule. Children were expected to do what their parents did and to become what their parents had been. The rule maintained the existing hierarchy of noblemen, clergy and farmers, and made life predictable with a fixed meaning:

the earthly hierarchy resembled the heavenly hierarchy presided by God (Marder 2018). While being from a noble heritage, Hildegard could have been predestined to become a noblewoman. However, her visions were difficult to understand for herself as well as for her parents or teachers. This made Hildegard "strange". She did not fit in with the regular scheme. For such persons, there was a predetermined path: they served the church or entered the monastery. So, it happened to Hildegard. She was gifted with something extraordinary, not understood very well, not fitting in with the society at that time. The solution was to hide and confine the extraordinary by giving her to the church.

It was a custom of medieval families to pay tribute to the catholic church by having one of the children enter the monastery or joining the clergy. There even was a rule for this: the "tithe". As Hildegard was the 10th child in her family, she was given to the church as a "tithe", since she represented 1/10th of the children produced. Girls became nuns. Hildegard did not become a "regular" nun but an anchoress. She was enclosed in a cell adjoining the church wall with an older religious woman named Jutta. The hiding theme reveals itself.

Hildegard hid her visions for many years and served the anchoress Jutta who taught her the ideas and rites of the church. She hid her being different, until one day it came out. Hildegard wrote about it as follows: "So arise and cry and say what is revealed to you by the most powerful agency of divine help. For He Who powerfully and benevolently commands every one of His creatures, flows with clear clarity of heavenly splendor all those who fear Him and who serve Him in sweet devotion and spirit of humility. And He leads those who persevere in the way of righteousness to the joy of eternal vision" (Hildegard of Bingen 1986, Part I, sct. 1). The extraordinary manifested itself in Hildegard, so pure and strong it could not be hidden anymore. She speaks of it as being in a flow and revealed her visions: "And again I heard a voice from heaven saying to me: So call and so write" (Hildegard of Bingen 1986, Part I, sct. 3).

The visions that she hid from others now became public. Such visions contained visual images accompanied by a voice: "And again I heard a voice from heaven, which spoke to me: God, who created all, made humanity in his own image and likeness, and in them he marked out both the higher and lower creatures. He had such love for humanity that he destined them to take the place from which the falling angel had been ejected, and he ordained them for the glory and honour which the angel in his bliss had lost. This is shown by the vision you see" (Hildegard of Bingen 2018, Part I, sct. 3). Hildegard's visions express a lesson, the moral of the story, explained by: "This signifies symbolically that things may be perceived through faith that cannot be seen visibly with the eyes" (Hildegard of Bingen 2018, Part I, sct. 3). So, her visual images are abstract by nature and expressed by a voice incorporated by Hildegard. The explanation was in sync with the experience. The visions became manifest in a withdrawal from the world but did not remain secluded, as Hildegard did not become the anchoress she was destined to be. As a counselor, we meet a first dialectic in her life (hiding versus expressing) that points out a theme to be further explored by the vertical axis of the meaning system: the relationship of Hildegard with "The Higher".

## 4. Hildegard and Her Relationship with God

The medieval monastery was a secluded place for serving God by keeping the ancient rites and scripts. It was a place where personality was given up and one dressed in a habit, took up another name in honor of a saint and followed an age-old tradition of never changing daily rites (the tides) and services. It was not a place for standing out or developing personal objectives. The character of the medieval monastery and her extraordinary endowment brought Hildegard another dialectic in her life (being no one versus being famous).

In the monastery of Disibodenberg, Hildegard struggled with her position. Her relationship with God was fundamental. Hildegard wrote: "That very bright fire you see refers to the almighty and living God. In His exceedingly clear brightness He has never

been touched by any iniquity. He remains incomprehensible, for no division can divide Him. He knows neither beginning nor end, and no spark of knowledge of His creature is able to comprehend what He is like. He cannot be wiped away, for He is the fullness that finiteness has never touched. Absolutely nothing is hidden from Him that He would not know. He Himself is all life, for everything that lives receives life from Him" (Hildegard of Bingen 1986, Part II, sct. 1).[3] However the monastery of Disibodenberg could not provide a form for this kind of relationship. Hildegard's visions appealed to many people and made her famous, but the monastery was no place for famous people. Again, Hildegard did not fit in. Her relationship with God being determined by the visions supervened her relationship with the structure of the monastery. Hildegard broke out and founded her own monastery. As a counselor, we can conclude the vertical axis was not only defined by her relationship to God, but also by autonomy as value serving her relationship with God. Hildegard was endowed with a strong will to establish a relationship with God according to the content of her visions. For this, freedom from traditional restraints was needed. In her own monastery Hildegard could build a relationship with God, while being herself. This calls for an exploration of the reflexive axis in the system of meaning: the relationship of a person with herself.

## 5. Hildegard and Her Relationship with Herself

In the monastery of Rupertsberg, Hildegard could finally be what she wanted to be. She could express her visions in writings, a mystery play, and music. She transcended herself in visions and worked towards a harmonious state best exemplified by her music. Harmony is her theme now. Hildegard wrote: "And again I heard Him say to me, "O how beautiful are your eyes in the story of God, for there the dawn shines according to God's decree." And again I replied, enlightened by what I understood of the inward insight of that vision: In the depths of my soul I appear to myself as ashes of ashes of rubbish and as billowing dust, in which I sit trembling like a feather in the shadows. Destroy and yet do not remove me from the earth of the living, for in this vision I work up a sweat" (Hildegard of Bingen 1986, Part III, sct. 2). Hildegard is herself, while dissolving in God. The image of the ash and sweat is picturing her as material on the one hand, but ethereal as ash and sweat can be on the other. The relationship with herself is a dialectic of matter and spirit. It is also a dialectic of (a) spirit and (the) Spirit. In her letter to Bernard of Clairvaux, Hildegard writes: "I am very concerned by this vision which has appeared to me in the spirit of mystery, for I have never seen it with the external eyes of the flesh. I who am miserable and more than miserable in my womanly existence have seen great wonders since I was a child. And my tongue could not express them, if God's Spirit did not teach me to believe" (Baird and Ehrman 1995).

The struggle with these dialectics dissolves in a harmony expressed by music.[4] Hildegard wrote: "Then I saw a very bright shining sky. In a miraculous way I heard there, in all the keys already described, different kinds of music. They were the praises of men who were full of exalted joys and who had persevered strongly in the way of truth. But there were also the lamentations of those who had been called back to praise those same expressions of joy, and who exhorted themselves to be virtuous for the good of the nations. Diabolical ambushes oppose this. But the virtues resist the ambushes, so that the believing people finally pass through penance from sin to the lofty realms. And that voice resounds as in a great chorus of a great crowd of people. Sing together in unity" (Hildegard of Bingen 1986, Part III, sct. 13). So, the third axis in the system of meaning can be conceptualized by "harmony", a concept which applies to Hildegard's way of being in an ontological and in a practical sense.

## 6. A Spiritual Exercise

A further elaboration on harmony serves a deeper understanding of Hildegard. In philosophical practice, we use exercises for this (Weiss 2015). This might seem strange to "theoretical" readers, but it exemplifies philosophical practice as that part of philosophy

which aims at giving participants a philosophical experience (Achenbach 2010). As a practice is a kind of common (reflexive) action (Foucault 1989), the exercise is actually a group exercise. Answers are discussed and the experience is shared with others. Such a setting is not possible while reading, but let us attempt to draw the reader of this paper into the following exercise I developed in order to make you think like Hildegard. Take each step (without looking at the next) after each other and write your answers down.[5]

1.  First listen to: https://www.youtube.com/watch?v=7G0dsrjQmAU[6]
2.  Then in silence, read these citations out of Hildegard's work carefully:

    a.  "And so, once liberated, man shines in God and God in man. Because man is now a partner of God, he now shines more intensely than before when he was still in heaven" (Hildegard of Bingen 1986, Part I, sct. 31)
    b.  "In this vision my soul, as God would have it, rises up high into the vault of heaven and into the changing sky and spreads itself out among different peoples, although they are far away from me in distant lands and places" (Letter of Hildegard to Guibert de Gembloux, Storch and Lauer 1993).
    c.  "They were all like flaming torches moved by a flying wind. This (wind) is full of voices that sound like the sound of the sea, and, that wind in anger raises its voices, and he sent fire into the darkness of the cloud, which kindled it in its darkness, but without flames" (Hildegard of Bingen 2018, Part III, sct. 1).

    Answer the question: Which citation a, b, or c suits the music you have heard *best*? (Choose one! You have to decide. Be guided by the normative: best)
3.  Explain *why* you think the citation chosen fits the music in the form of an argument, i.e., "I think the citation fits the music best, because..."
4.  Conceptualize, i.e., give a key word in or underlying your argument. (One word, composite or neologism allowed)
5.  Evaluate: What is the relationship between your concept and "harmony"?

## 7. Thinking like Hildegard

Thinking like Hildegard makes us understand Hildegard, not in a theoretical, but in a practical way.[7] This was the purpose of the exercise the reader was invited to do. Although we cannot reflect on the experience of a particular reader here, a reflection on the didactical and philosophical principles of the exercise will bring us closer to an understanding of Hildegard.

The structure of the exercise I developed captures, first of all, the basic elements of a philosophical life: choice, commitment to the choice (take a stance), argument, conceptualization and evaluation—a reflexive process characterizing philosophy put into practice. It involves philosophical competences such as phenomenological description of an experience, reasoning and questioning. Philosophical practice acknowledges the general presence of these competences as part of the human condition, more or less developed in particular human beings and exemplified by the (academically trained) philosopher. A structured exercise enables a person not trained in academic philosophy to practice and develop philosophical competences.

The exercise starts with listening to a contemporary remix of Hildegard's composition called "Vision". Visions are a major theme in her life. The music first invites the listener to a phenomenological description of the experience. Hildegard's music is monophonic, that is, consisting of exactly one melodic line (Burkholder et al. 2010). The repetition of this line enables transcendence and creates a spiritual atmosphere. This suits the rhythm of the mellow house beat perfectly. The mix is an interplay of a powerful beat and the fragile voice on a spacious melody. The mix also enables a connection between the ages. Rhythm and space carry the listener back and forth between their world and Hildegard's. The line "O Euchari, in leta via ambulasti" (Eucharius, you walked a joyous road) is repeated in the contemporary remix as exaltation and qualification of the way to go.

The music is the object of conceptualization.[8] Three citations are offered, expressing different aspects of Hildegard's thoughts. The assignment to choose one of the citations suiting the music best is intended to conceptualize the experience of the listener. The citations also serve the understanding of Hildegard not only by her music but by her words (text) as well. The citations exemplify the different stages of Hildegard's life (para. 3–5): citation a is about the relationship of Hildegard with the world, citation b is about Hildegard's relationship with God, and citation c is about the relationship with herself. The participant is invited to choose the one that *best* fits the music. Of course, there is no good answer. All citations might fit. However, the particular choice made is intended to make a person reflect on who (s)he is. In philosophical practice, the identity of a person is involved in the philosophizing. The first quote (a) is usually chosen by people internalizing, but shining from freedom, the second (b) by people dissolving in the rhythm or space of the music, the third (c) by people externalizing power. The choice made can be used by the philosophical counselor to analyze the participant and bring her closer to herself. By stressing the search for the *best* fit, the participant's (personal) normative frame of reference is addressed. It becomes explicit in her choice and is the object of further exploration by the argument and underlying concept.

The Socratic style of philosophizing is brought into the exercise by asking an argument for the citation chosen. For instance, an often-heard argument of people doing the exercise is: "I think citation x fits the music best, because it is *in* the music". The argument exemplifies the logic of the participant. By being connected to the musical experience, this is not a formal logic, but a lived logic that is being investigated. When well concentrated, the participant is *in* the music.

Conceptualization is the next step. A basic competence is addressed that characterized philosophy through the ages: from Socrates to Hadot, philosophers brought up (new) concepts to understand themselves, the other or the world. Participants doing the exercise are invited to follow these examples, as philosophizing is not restricted to the professional philosophers. Participants came up with, for example, concepts such as: freedom, energy, space, warmth, movement, upheaval, rhythm, power and enlightenment. Because the concept stems from a choice of the *best* fit, the concept resembles the participant's normative frame of reference on the one hand and relates to the content of the citation on the other hand.

In the evaluation, the participant is led back to the world of Hildegard by asking about the relationship between their concept and "harmony". Participants doing the exercise actually then defined harmony by use of their (personal) concept, e.g., "harmony is balanced freedom", "harmony is warm energy", etc. The participant's normative frame of reference is connected with the experience, deepens it and turns it into a practice by the connection with a (general) concept (Foucault 1989). The normative framework is not used to judge that experience as in ordinary live, but to evaluate it and to transcend it via conceptualization, so that the participant will have a spiritual experience, which is in accordance with the object of the exercise, i.e., Hildegard's music.[9]

## 8. Understanding Hildegard

Attempts to understand Hildegard leave most interpreters confused. Hildegard is hard to classify by the tendencies of her time. With retrospect, she is seen as a sage (although uneducated), a feminist, a soothsayer, a doctor and a saint. She is called a rolling stone in history at the etch of a period of symbolism and Neoplatonism that would soon be replaced by the sober analytic approach of Aristotle (Hildegard of Bingen 2000). Hildegard's major themes—visions and her frequent periods of illness—inevitably led to medical interpretations of her life. In the early 20th Century, Singer (2005) suggests a medical explanation for Hildegard's visions as "migraine accompanying phenomena". Oliver Sacks (1985) later followed and spread this interpretation in his popular books on mental phenomena. However, as a medical doctor, I think that her visions cannot be classified as pathological. There are voices accompanying the images. Migraines are not accompanied

by voices. According to the DSM-V, the current manual for psychiatric diseases (APA 2022), hearing voices that instruct behavior are a hallmark of schizophrenia and seeing images are a hallmark of delirium. An interweaving of voices and images cannot be classified as a delirium or symptom of schizophrenia. Also, the voices and images are followed by thoughts as reflection on the experience. This looks like a kind of meditation, a mental exercise for reaching a higher level of spiritual awareness (Hadot 1995).[10] In the case of Hildegard, visions serve a deeper understanding of the world, the gospel (God's word) and herself. They are transcendental in the sense that the person of Hildegard dissolves in the vision. So, Hildegard's visions are a kind of transcendental meditation rather than pathological phenomena. Medical interpretations should be avoided. They suffer from change and progress in medical knowledge and can only offer contingent explanations that seem to serve the societal tendencies of the corresponding time. What once could be called "migraine accompagnée" does not fit current pathological criteria anymore and becomes outdated.

Werthmann (1993) offers a psychoanalytic approach to Hildegard's life. She does not focus on the usual, but on what deviates from it. By paying attention to Hildegard's own way of expressing herself, she said she was able to distill tensions resulting from personal conflicts from various passages. She also considered it possible to read variations in religious themes as personal views that are comparable to the free association in psychoanalysis. Werthmann pointed out several traumatic events in the course of Hildegard's life and discovered the ingredients of a narcissistic coping process. In line with psychoanalytical theory, she explained Hildegard's visions as a withdrawal into a closed inner world.

While the academic approach aims at interpreting the works of Hildegard, philosophical practice aims at understanding the life of Hildegard, not as a biographer would do from a historical point of view but as practitioner from a philosophical point of view. Such an approach was initiated by the French philosopher Pierre Hadot (1995) as he outlined the *figure* of Socrates and Marcus Aurelius.[11] A figure incorporates an understanding of the meaning of a life with regard to a contemporary context in the character of a person. A figure results from a study of biographical facts, original text fragments (the words) and *possible* interpretations of the person as a claim not to truth, but to meaning. The historian would call such an approach too speculative and the academic philosopher too practical.[12] However, it is the acknowledgement of the fact that we cannot escape contemporary context in interpreting a historical person. For example, the Socratic method is not to be found in the works of Plato as such, but a reconstruction out of elements of the life and work of Socrates serving contemporary purposes (education, empowerment, analysis, group processes, etc.). In the paradigm of philosophical practice, it suits the understanding of both historical as well as (actual) contemporary persons, i.e., clients of a philosophical practice. It also serves the purpose of further developing the experience (Ger. "Erweiterung"), not closing it in or even destructing it by analysis.

In philosophical practice, the understanding of a person is not led by a particular theory, but rather moves from form to content (Harteloh 2023b). The philosophical practitioner understands the counselee, and the counselee understands the philosophical practitioner in a reflexive process so that eventually the counselee will be able to understand herself. In a philosophical consultation, we look at the life of a person, search for the right questions, listen to words, interpret them in a philosophical and systematic way and construct a metaphor in order to capture meaning (Harteloh 2013). In philosophical practice, a citation, a story and scientific theory all serve as metaphors to capture meaning.

Hildegard herself captured the meaning of her life in 2 Corinthians 12:9: "My grace is sufficient for thee, for My strength is made perfect in weakness", a recurring citation in autobiographical fragments. The citation fits well, as it suits a life in which meaning is sought in a connection of personal experience with religious truth. Captured is a theme: visions of absolute wisdom in relation to severe physical illness. It is also an expression of Hildegard's morale: be small and modest in order to open up for God. Weakness becomes strength. A dialectic between two opposites. The form underlines the dialectic

development of her life. The philosophical (and not theological) content is found by the connecting quality carrying the logical oppositions in real life. In the case of Hildegard, I would propose "energy", in line with Aristotle's theory of the mean. Strength and weakness are qualities of energy. Energy can be limited (weakness) or fully developed (strength). A controlled energy is the mean. This is (finally) found at the Rupertsberg. The life of Hildegard appears as a search for the right place. She moved from Böckelheim to Disibodenberg to end up in Rupertsberg, a place where she could be herself. It is a finely placed site high in the hills. Between the little settlement and the important medieval town of Bingen flowed the River Nahe. The stream was spanned by a bridge of Roman origin, to which still clings the name of the pagan Drusus (15 BCE to 19 CE). Thus, Hildegard ended up in a place of her own, secluded and yet linked to the world, where she could live and express her visions. However, in her visions Hildegard saw things as they actually are, an eternal state of affairs not bound to any place whatsoever. She transcended the Rupertsberg by her visions.

The imaginary philosophical consultation shows Hildegard as being driven by a search for a "topos"—the ancient Greek word for "place" in a physical sense, but also in a metaphorical sense, i.e., a topic. This opens up a final dialectic. The visions are her place (topic), but in those visions Hildegard was not in a particular place anymore. This not belonging to a particular place haunted her from early on, resulting first in a search for the right physical place to be and in the end to a transcendental state of harmony not attached to any place whatsoever. The French philosopher Pierre Hadot (1995, p. 158) (re)discovered the Greek concept "átopos", literally "out-of-place-ness" for designating such a kind of life. The term indicates someone being strange, extravagant, absurd or unclassifiable, terms that apply to the life phases of Hildegard of Bingen too: strange in her youth, considered extravagant or absurd as a seer in her middle life and unclassifiable in the end. Hadot also discovered "átopos" as designating the character of a *philosophical* life: "It is the love of this wisdom, which is foreign to the world, that makes the philosopher a stranger in it" (Hadot 1995, p. 57), and "Now we have a better understanding of átopia, the strangeness of the philosopher in the human world. One does not know how to classify him, for he is neither a sage nor a man like other men. He knows that the normal, natural state of men should be wisdom, for wisdom is nothing more than the vision of things as they are, the vision of the cosmos as it is in the light of reason, and wisdom is also nothing more than the mode of being and living that should correspond to this vision" (Hadot 1995, p. 58). Thus, átopos is a concept connected with a way of being on the one hand and with a vision of wisdom on the other. It perfectly suits Hildegard's journey to harmony. A philosophical diagnosis can be put on her theme now. Hildegard's life is a philosophical life! The moral of the story or the lesson to be learned is that such a life can be achieved by spirituality. Such a moral is not a theory but expressed by Hildegard's music as vehicle of spirituality. The understanding of Hildegard lies in the experience of her music. This music enables the listener to transcend herself, enter the world of Hildegard and move into a state of absolute harmony.

## 9. Conclusions

In philosophical practice, a conclusion is not a *definite* answer, i.e., a final claim to truth. Such a claim may be the purpose of science, but it does not belong to philosophical practice. Philosophy is considered a process without end. It can best be pictured as a cycle or spiral, an ever-continuing, changing process, deepening itself during the course of life. In that process, there are lessons to be learned, lessons as a temporary point of rest in spacetime, to be generalized as rules of thumb suiting life or as the moral of the story (Montaigne 1965). The general lesson to be learned from consulting Hildegard is as follows: spirituality means transcending yourself by being átopos. The personal lesson to be learned from the exercise proposed in this paper is to the reader.

**Funding:** This research received no external funding.

**Institutional Review Board Statement:** Not applicable.

**Informed Consent Statement:** Not applicable.

**Data Availability Statement:** No new data were created or analyzed in this study. Data sharing is not applicable to this article.

**Conflicts of Interest:** The author declares no conflict of interest.

## Notes

[1]  For this, I will use Hildegard's ideas laid down in three major works: "Scivias" or "knowing the way" (written between 1141 and 1150), the "Liber vitae meritorum" or "Book of Life's Merits" (written between 1158 and 1162), and the "Liber divinorum" or "Book of Divine Works" (written between 1163 and 1170) (Hildegard of Bingen 1986, 1995, 2018). The "Vita van Hildegard" (life of Hildegard) is a source of (some) autobiographical remarks (Hildegard of Bingen 2000). She also wrote a mystery play (Ekdahl Davidson 1985), developed her own language (Higley 2007) and composed music (Burkholder et al. 2010), incorporating the practical application of her ideas. In this paper, I will use her music to meet Hildegard in philosophical practice.

[2]  My intention is not to deny theological interpretations of Hildegard's life, for instance, to see her life as fulfilment of God's will or to consider Hildegard as a saint. I disregard these interpretations here from a methodological point of view in order to add something to the corpus of understanding from a philosophical point of view in line with Anselmus (2000): *fides quaerens intellectum.*

[3]  In terms of Paterology, Hildegard considers God as creator and protector, but above all her longing for a unity with God defines her spirituality as exemplified by her visions (Soto-Bruna 2023).

[4]  Although, it also fed an anti-dialectical position that she assumes—along with Bernard of Clairvaux and others—making her an opponent of the novelty of the cathedral schools and a defender of the early medieval paradigm.

[5]  As the exercise is intended to be done in a group or with a counsellor, here it can only be considered an example. However, the reader is invited to correspond about her experiences via Harteloh.

[6]  Accessed on 14 April 2024. In case the link does not work (anymore): The Music Of Hildegard Von Bingen 12″ Promo, Translucent Blue Vinyl, Angel Records—SPRO 79985 (1994).

[7]  Philosophical practice is that part of philosophy in which it is not the intention to think *about* philosophers, but to think *like* the philosopher under consideration (Hadot 1995). Exercises play a crucial role in this kind of philosophy.

[8]  In accordance with the trend that nowadays modern people seem to reach spirituality much easier by music than by reading texts.

[9]  After Kant (2007): an experience without concept is blind, but a concept without experience is blind too.

[10]  Understanding Hildegard's visionary experiences as mental exercises is not intended to deny the theological interpretation of their nature. The visionary phenomenon as presented in Hildegard is not an exercise that obeys her will, but the one who takes the initiative is God. Hildegard is, then, a "chosen one" of God, so to speak. She does not choose to have the visionary experience. The way she is dealing with this experiences is a philosophical exercise practiced until a state of harmony is reached. The present paper focuses on this.

[11]  Also, a religious practice can be considered a kind of philosophical practice (Sharpe 2023).

[12]  In line with the original character of philosophy as a folk phenomenon, functioning outside academy at the market place, including debate, rational thinking and speculation, before it was incorporated in academy as "scientific" discipline in the late 18th Century (Hadot 1995).

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
