# Peer review of "Hildegard of Bingen: Philosophical Life and Spirituality"

_religions, doi:10.3390/rel15040506_

Round 1

Reviewer 1 Report

Comments and Suggestions for Authors

Commentary:

The article tries to draw lessons from Hildegard's life, from the 12th century, that can be used in a philosophical consultation in the 21st century, although this runs the risk of "modernizing" her like the contemporary remix Vision modernizes her music. It discusses Hildegard's relationships with the world around her, with God (the Highest) and with herself. It adds a spiritual exercise usable in philosophical consultation and considers that it is possible to understand Hildegard and think like her from her music. Hildegard's is a philosophical life motivated by the search for a place (topos) that she finds in a harmony that the author, based on P. Hadot (1995), interprets as an atopos, an “out-of-place-ness.”

Although the use of Hildegard in philosophical counseling is always interesting, I would like to clarify some issues according to my particular point of view:

1. I think that Hildegard is not looking for a place where she can finally be herself (in a modern existential sense), but where she can fulfill God's will. She did not come to Rupertsberg to be what she wanted to be (l. 166), but to be what God wanted her to be.

2. In the l. 161 seems to draw a Hildegard who opposes tradition. This is not false in some sense, but it should be clarified that the anti-dialectical position that she assumes – along with Bernard of Clairvaux and others – makes her an opponent of the novelty of the cathedral schools and a defender of the early medieval paradigm.

3. I maintain that in order to think like Hildegard – and understand her music (l. 396) – it is also necessary to understand her writings, since they are not alien to her life.

4. Understanding Hildegard's visionary experiences as mental exercises whose object is to reach a higher level of spiritual awareness (l. 307-308) can obscure the visionary phenomenon as presented in Hildegard, which is not an exercise that obeys her will, but the one who takes the initiative is God. Hildegard is, then, a “chosen one” of God, so to speak. She does not choose to have the visionary experience.

5. Although in the vision she seems to place herself on a different spiritual plane while she is sober and aware of the physical place in which she is, I think that the place she is looking for is not precisely the plane of the vision, since she does not seek the vision for its own will but receives it through divine intervention. Hildegardian anthropology, which places man not in an atopos but in the very center of the earth and crossed by all cosmic forces, would add a necessary nuance to the description of the Hildegardian locus. This, together with the defense of ora et labora against the liberal arts of the magistri, distances the idea that Hildegard seeks to “fly from the world.”

6. Hildegard is a teacher for the 21st century not because she coincides in certain existential aspects with the interests of this century, but precisely because she proposes an alternative paradigm to the dominant modern paradigm.

Corrections:

l. 35: on the one other hand.

l. 96: noblewoman.

l. 103-104: the sentence seems to be incorrectly constructed.

In l. 84 it’s read “The Higher” (uppercase) but in l. 135 “the higher” (lowercase).

l. 206: I find it more appropriate to write the title and the reference of the music in the body of the text and then in a footnote to inform that it can be heard in the YouTube link. Otherwise, the video could be deleted and the reader would not even have the title and album to be able to search for it elsewhere. The Music Of Hildegard Von Bingen 12"  Promo, Translucent Blue Vinyl, Angel Records ‎– SPRO 79985 (1994).

l. 296: correct the citation (Lindijer 2000) (Von Bingen 2000).

l. 363: Desibondenberg Disibodenberg.

l. 403: philosophy is a process, without end. (delete the comma).

l. 423-426: I don't understand the "Xxx" references.

Author Response

April 15, 2024

Dear editor,

First of all, I would like to thank the three anonymous referees for their profound comments and correction of typos. They recognized, but did not reject the difference between a theological and a philosophical approach and by their comments I could improve the manuscript.

I adjusted the manuscript accordingly:

Referee 1.

I thank this referee (number 1) for his/her profound comments. They seem to stem from a theological point of view with which I cannot but agree. However, the intention of this paper differs. I paid attention to the points mentioned as follows:

  1. I think that Hildegard is not looking for a place where she can finally be herself (in a modern existential sense), but where she can fulfill God's will. She did not come to Rupertsberg to be what she wanted to be (l. 166), but to be what God wanted her to be.

Indeed, this is a (standard) theological explanation of the course of her life. In my paper, I approached her life from a(n) (alternative) philosophical point of view. However, I do not deny this theological interpretation. I added a footnote (new numbering 1) on this.

  1. In the l. 161 seems to draw a Hildegard who opposes tradition. This is not false in some sense, but it should be clarified that the anti-dialectical position that she assumes – along with Bernard of Clairvaux and others – makes her an opponent of the novelty of the cathedral schools and a defender of the early medieval paradigm.

True, footnote (new numbering 4) added.

  1. I maintain that in order to think like Hildegard – and understand her music (l. 396) – it is also necessary to understand her writings, since they are not alien to her life.

I agree. It was not my intention to disregard her writings, but I focused on her music as way of reaching spirituality (in accordance with the theme of the special issue). Mind, that texts of Hildegard are part of the exercise too. It is switching back and forth between music and text. Also, for people nowadays it seems easier to reach spirituality by music than by reading (Unfortunately!). Added a sentence to make this clear. 

  1. Understanding Hildegard's visionary experiences as mental exercises whose object is to reach a higher level of spiritual awareness (l. 307-308) can obscure the visionary phenomenon as presented in Hildegard, which is not an exercise that obeys her will, but the one who takes the initiative is God. Hildegard is, then, a “chosen one” of God, so to speak. She does not choose to have the visionary experience.

I (fully) agree. However, the way she is dealing with her visionary experiences belong to her (human) will and choice. For instance, she decided to hide them for others during early part of her life. See the biographical notes de Vita (Hildegard of Bingen, 2000). Present paper focus on this aspect of her life.  Footnote with new number 10 added to acknowledge this fact. I thank the reviewer for this remark.

  1. Although in the vision she seems to place herself on a different spiritual plane while she is sober and aware of the physical place in which she is, I think that the place she is looking for is not precisely the plane of the vision, since she does not seek the vision for its own will but receives it through divine intervention. Hildegardian anthropology, which places man not in an atoposbut in the very center of the earth and crossed by all cosmic forces, would add a necessary nuance to the description of the Hildegardian locus. This, together with the defense of ora et laboraagainst the liberal arts of the magistri, distances the idea that Hildegard seeks to “fly from the world.”

Good point. I wonder in what way Hildegard identifies herself with the position of men in her visions. Probably she does. But this position is not located in space-time.

  1. Hildegard is a teacher for the 21st century not because she coincides in certain existential aspects with the interests of this century, but precisely because she proposes an alternative paradigm to the dominant modern paradigm.

I agree when we focus on her writings. However, with regard to her music and spirituality I would claim Hildegard does coincide with modern existential aspects as the attraction of her music to contemporary listeners indicates.

The word “morale” has been revised where it was meant as “moral of the story”. 

Changes have been marked yellow.

The number of self-references has been reduced.

The typos have been corrected. I thank the referees for noticing them.

The manuscript has undergone a minor revision. I hope it can be published now.

Kind regards,

Reviewer 2 Report

Comments and Suggestions for Authors

The article is interesting, especially regarding Hildegard of Bingen's relationship with herself and with God. It would be good if he compared his works with Latin Patrology and Corpus Cristianorum.

Is very important the studies of Claudia D'Amico and M. J. Soto-Bruna.

Author Response

April 15, 2024

Dear editor,

First of all, I would like to thank the three anonymous referees for their profound comments and correction of typos. They recognized, but did not reject the difference between a theological and a philosophical approach and by their comments I could improve the manuscript.

I adjusted the manuscript accordingly:

The word “morale” has been revised where it was meant as “moral of the story”. 

Referee 2. Is very important the studies of Claudia D'Amico and M. J. Soto-Bruna.

Unfortunately, I do not see works of Claudia D'Amico in English. Incorporated a reference to Soto-Bruna.

Compared his works with Latin Paterology and Corpus Cristianorum.

Added a footnote on this and two references which certainly enrich the paper. I thank the referee for that.

Soto-Bruna, Maria Jesús. Religious Vocabulary on Creation: Eriugena, Hildegard of Bingen, Eckhart.

Religions 2023, 14(8), 1024; https://doi.org/10.3390/rel14081024

Lerius, Julia. 2018. Hildegard von Bingen on Autonomy. In: Sandrine Bergès and Alberto Siani (ed.). Women Philosophers on Autonomy: Historical and Contemporary Perspectives. Oxfordshire: Routledge, Taylor & Francis Group, Chapter 2.

The word “morale” has been revised where it was meant as “moral of the story”. 

Changes have been marked yellow.

The number of self-references has been reduced.

The typos have been corrected. I thank the referees for noticing them.

The manuscript has undergone a minor revision. I hope it can be published now.

Kind regards,

Reviewer 3 Report

Comments and Suggestions for Authors

Hildegard von Bingen: Philosophical Life and Spirituality

 Hildegard of Bingen is a figure who appeals through her music and visual way of thinking to many readers outside a formally academic milieu. This is a paper that goes beyond normal academic convention through appealing to Hadot’s notion of philosophy as a way of life (exercice spirituel in the original French version of his classic text. Hildegard herself might not have used the term philosophical life, but the author’s intent is clear: to suggest ways in which Hildegard’s music and reflections can be used to initiate contemporary reflection. I appreciated the way that this paper did not do violence to Hildegard’s thought or its context (as often happens with contemporary appropriation of it), but rather spells out a methodology for individuals who may not be scholars to benefit from her writing and music. There are many different ways in which this could be done. Perhaps this paper is rather thin on certain themes, like viriditas. Inevitably, it has to work through translations from Latin, rather than engaging in a philological way with the concepts that she used. But overall, I find this a fresh and original contribution.

My queries are just about some issues of translation into English. In particular, calling her Hildegard von Bingen implies that von Bingen is an aristocratic name, which is not the case. She is of Bingen, because this was her abbey not because of her family. I would suggest modifying both the title to Hildegard of Bingen and bibliographical references to (Hildegard, 1999). I have observed a practice among some English speaking devotees to refer to her as von Bingen as if this is her name, but this projects a pseudo-aristocratic name onto her (different to Jutta von Sponheim, which is a family name). Other corrections:

96  predistinated ] CHANGE TO predestined

103 Motivation was a custom… was to pay] problem with syntax. Better: It was a custom of medieval families to pay

107 Mark, Hildegard]   Mark does not work in English like this. Just omit.

138 It was a place where you gave up…]  Avoid you. It was a place where personality was given up. One dressed…  (but you is OK in line 223 etc)

153-5  the word famous is used four times in three lines. Improve prose style here.

312 the change] change

325 and 333 academical] academic

351 in 2. Corinthians] through 2 Corinthians

367 very old fashioned to use BC and AD. Say 15 BCE-19 CE.

392 the other hand.] the other.

400 Morale Is this a mistake for Moral?   Morale means sense of self-esteem, while Moral means a moral lesson to be drawn from a story. The word chosen here is never explained. I think it would be clearer to say Conclusion.  The final paragraph never uses or explains the word  Morale, and repeats the word conclusion twice In two sentences, not very good style (nor is beginning a sentence with However good style). It is still a conclusion to conclude that there is no final answer.  I’d suggest changing the second sentence to: In philosophical practice, however, there is no definite answer…

Comments on the Quality of English Language

English generally very good, with just a few issues needing attention

Author Response

April 15, 2024

Dear editor,

First of all, I would like to thank the three anonymous referees for their profound comments and correction of typos. They recognized, but did not reject the difference between a theological and a philosophical approach and by their comments I could improve the manuscript.

I adjusted the manuscript accordingly:

Referee 3. Von Bingen or of Bingen? Indeed a very good point. In German von can mean both: being of noble heritage or being from a particular town. In standard English it should be Hildegard of Bingen. I followed the referee and changed Hildegard von Bingen to Hildegard of Bingen in text and references. In referencing to her work I followed Soto-Bruna, Religions 2023.  

The word “morale” has been revised where it was meant as “moral of the story”. 

Changes have been marked yellow.

The number of self-references has been reduced.

The typos have been corrected. I thank the referees for noticing them.

The manuscript has undergone a minor revision. I hope it can be published now.

Kind regards,